# Racial Disparity in Quadruple Negative Breast Cancer: Aggressive Biology and Potential Therapeutic Targeting and Prevention

**DOI:** 10.3390/cancers14184484

**Published:** 2022-09-16

**Authors:** Nikita Jinna, Tijana Jovanovic-Talisman, Mark LaBarge, Rama Natarajan, Rick Kittles, Christopher Sistrunk, Padmashree Rida, Victoria L. Seewaldt

**Affiliations:** 1City of Hope Comprehensive Cancer Center, Duarte, CA 91010, USA; 2Beckman Research Institute of City of Hope, Duarte, CA 91010, USA; 3Novazoi Theranostics, Salt Lake City, UT 84105, USA

**Keywords:** quadruple negative breast cancer, triple-negative breast cancer, androgen receptor, African-American, racial disparity, epigenetic modifications

## Abstract

**Simple Summary:**

Quadruple negative breast cancer (QNBC), a subgroup of triple negative BC, has emerged as a highly aggressive BC subtype that disproportionately afflicts and impacts Black/African-American (AA) women. In this article, we review molecular distinctions in Black/AA and White/European-American (EA) QNBC biology as well as address potential non-genetic risk factors that could be underlying this racially disparate burden. We aim to provide deeper insight and provide a framework for novel discovery of actionable therapeutic targets and identify lifestyle changes to improve outcomes for Black/AA QNBC patients.

**Abstract:**

Black/African-American (AA) women, relative to their White/European-American (EA) counterparts, experience disproportionately high breast cancer mortality. Central to this survival disparity, Black/AA women have an unequal burden of aggressive breast cancer subtypes, such as triple-negative breast cancer (ER/PR-, HER2-wild type; TNBC). While TNBC has been well characterized, recent studies have identified a highly aggressive androgen receptor (AR)-negative subtype of TNBC, quadruple-negative breast cancer (ER/PR-, HER2-wildtype, AR-; QNBC). Similar to TNBC, QNBC disproportionately impacts Black/AA women and likely plays an important role in the breast cancer survival disparities experienced by Black/AA women. Here, we discuss the racial disparities of QNBC and molecular signaling pathways that may contribute to the aggressive biology of QNBC in Black/AA women. Our immediate goal is to spotlight potential prevention and therapeutic targets for Black/AA QNBC; ultimately our goal is to provide greater insight into reducing the breast cancer survival burden experienced by Black/AA women.

## 1. Introduction

In the United States (US), breast cancer is the leading cancer diagnosis and the second cause of cancer-related death in women [1]. Breast cancer is a heterogeneous disease that can be subdivided into 4 major intrinsic molecular subtypes—either by immunohistochemical (IHC) staining or PAM50 gene expression profiling [2]. The four breast cancer subtypes are: (1) luminal A (ER^+^/PR^+^/HER^−^), (2) luminal B (ER^+^/PR^+/−^/HER2^+^), (3) HER2-enriched (ER^−^/PR^−^/HER2^+^), and triple-negative breast cancer (ER^−^/PR^−^/HER2^−^, TNBC); these breast cancer subtypes are used to identify targeted therapeutic treatment and potential prevention options.

Breast cancer incidence and mortality rates differ significantly by race/ethnicity in the US. Black/African-American (AA) and White/European-American (EA) experience notably higher incidence and mortality rates of female breast cancer across all ages compared to American Indian/Alaskan Native (AI/AN), Hispanic, and Asian Pacific Islander (API) subpopulations, with rates being the lowest among APIs [3,4]. These racial/ethnic differences have been suggested to be primarily driven by higher rates of luminal A and B molecular subtypes observed among AI/AN, Hispanic, and API women but higher rates of TNBC observed among Black/AA and White/EA women [3]. Furthermore, White/EA and API women display the highest rates of localized breast cancers (64–66%) but lowest rates of regional stage breast cancer (27–30%). Whereas Black/AA and Hispanic women display the lowest rates of localized disease (56–60%) but highest rates of regional disease (35%) [4]. Distant-stage (metastatic) breast cancer contributes to 8% of diagnoses in Black/AA women compared to only 5–6% of diagnoses reported among other racial/ethnicities [4].

Among Black/AA and White/EA women, breast cancer disproportionately impacts Black/AA patients. Although the incidence rates between Black/AA and White/EA women are similar (126.7 vs. 130.8 per 100,000, respectively), Black/AA women experience a 40% higher death rate than White/EA women (28.4 vs. 20.3 per 100,000, respectively). Black/AA women are significantly more likely to present clinically with aggressive breast cancer subtypes such as TNBC, that lack an effective targeted therapy [3,5,6]. Black/AA women are twice as likely to be diagnosed with TNBC than White/EA women (38 vs. 19 per 100,000, respectively). In contrast, White/EA women are more likely to present with the least aggressive breast cancer subtypes, particularly luminal A breast cancer, that is effectively targeted with current therapies [3,4]. Thus, Black/AA women have fewer targeted treatment options compared to White/EA women, which has been suggested to underlie the racially disparate burden in breast cancer. Additionally, among all breast cancer subtypes, Black/AA women have the highest rate of recurrence and the lowest survival [5,6,7]. Within TNBC, even after adjusting for age, stage, grade, and poverty index, Black/AA patients experience significantly shorter survival (HR = 2.1, 95% CI: 1.1–4.0) compared to White/EA patients [8].

Recently a new molecular TNBC subtype has been identified —quadruple-negative breast cancer (QNBC) [9,10,11,12]. Similar to TNBC, QNBC lacks expression of ER/PR and does not overexpress HER2. In addition, QNBC lacks expression of the androgen receptor (AR) [13]. The absence of AR expression in TNBC is linked to more aggressive clinical features upon presentation, younger age at diagnosis, and shorter disease-free survival. QNBC more frequently occurs in Black/AA women (relative to White/EA women) and is emerging as a highly aggressive breast cancer subtype [10,13,14,15,16].

In this review, we examine and survey the current landscape of the racial health disparity in QNBC and review unique molecular features that distinguish racially distinct QNBC tumors. We aim to encourage the discovery of novel targets for therapeutic intervention in QNBC. Since QNBC frequently impacts Black/AA women and is highly aggressive, improving the therapeutic targeting and prevention of QNBC has the potential to substantially reduce Black/AA breast cancer disparities.

## 2. TNBC—The Triple Threat

TNBC is frequently referred to as a “triple threat” due to the absence of all three major therapeutic breast cancer targets—ER, PR, and HER2. [17,18]. In addition, TNBC is inherently more clinically aggressive than the other breast cancer subtypes as evidenced by the higher frequency of metastasis and recurrence within 5 years of diagnosis [17,19,20]. Linked with these poor survival statistics, TNBC is characterized by the highest inter-patient and intra-tumoral heterogeneity among the breast cancer subtypes [21,22]. Multiple groups have dissected the heterogeneity of TNBC starting with Lehmann and colleagues, who subdivided TNBC into six distinct molecular subtypes [23,24]. These subtypes include two basal-like subtypes (BL1 and BL2), immunomodulatory (IM), mesenchymal (M), mesenchymal stem–like (MSL) and luminal androgen receptor (LAR). Liu et al., recently integrated both long coding RNAs and mRNAs to classify TNBCs into 4 distinct subtypes, (1) immunomodulatory (IM)—enriched with immune cell and cytokine signaling; (2) luminal androgen receptor (LAR)—enriched with AR signaling; (3) mesenchymal-like subtype (MES)—enriched with growth factor signaling, and (4) basal-like immunosuppressed (BLIS)—enriched with cell cycle and DNA repair processes and downregulated immune response [25].

TNBC subtypes differ in their aggressive biological potential. The two basal-like subtypes (BL1 and BL2 or BLIS) and immunomodulatory (IM) subtypes carry a worse prognosis, while the AR+ LAR subtype carries a more favorable prognosis. Black/AA women more frequently have aggressive TNBC subtypes (BL1, BL2, and IM) and White/EA women more frequently present with the less aggressive LAR subtype [7,10,26].

## 3. AR Signaling/Pathway

AR is a type I nuclear receptor that is expressed in multiple tissue types in both sexes, including in the breast [27]. Although widely known to be instrumental in male biology as ER is in female biology, AR also plays a critical role in female biology [28]. AR is indispensable for both female fertility and breast growth [29]. The androgen, testosterone, is synthesized in the ovaries and adrenal glands in women and is converted to dihydrotestosterone (DHT) or 17β-estradiol (E2) in breast tissue. DHT or E2 binds to the AR or ERα to stimulate or inhibit cell proliferation [30,31,32]. When AR is not bound to its ligand, it is located in the cytoplasm, bound to heat shock proteins. Upon ligand binding, AR undergoes a conformational change, disassociates from heat shock proteins, becomes activated, and dimerizes with another activated AR [27]. These AR dimers translocate to the nucleus to bind to androgen-responsive elements (AREs) within target genes to modulate transcription. This AR-mediated gene transcription can result in differentiation, proliferation, apoptosis, or angiogenesis [33]. AR can also be activated independent of its ligand via crosstalk with key signaling pathways such as PI3K/Akt, ERK, mammalian target of rapamycin (mTOR), Wnt/β-catenin or via interaction with FOXA1 [34].

## 4. AR Pathway as a Therapeutic Target for TNBC

Nuclear AR is expressed in approximately 12–35% of TNBCs and has emerged as a promising therapeutic target [35]. AR inhibitors and antagonists, such as enzalutamide and bicalutamide, have elicited a promising response in vitro and in clinical testing. LAR TNBC cell lines had a higher sensitivity to bicalutamide [23]. In AR-positive TNBC models, both in vitro and in vivo, enzalutamide reduced proliferation, blocked invasion, and increased apoptosis [36,37,38]. In women with metastatic AR-positive TNBC treated with enzalutamide, in a nonrandomized phase II clinical trial, Traina and colleagues observed a clinical response rate of 25% at 24 weeks and a median progression-free survival (PFS) of 14.7 weeks to [39]. Similarly, in AR-positive metastatic TNBC, bicalutamide elicited a clinical benefit rate of 19% at 24 weeks and a median PFS of 12 weeks [40].

Apalutamide is structurally similar to enzalutamide but does not induce AR nuclear translocation or DNA binding [41]. Darolutamide antagonizes AR mutants such as F876 L, W741L, and T877A [42]. Apalutamide and darolutamide are currently under evaluation as promising new-generation AR inhibitors in phase III clinical trials for non-metastatic castration-resistant prostate cancer (NCT01946204 and NCT02200614, respectively) [43]. Thus, these new AR inhibitors may be tested in AR-positive TNBC patients in the future. Agents that target intracrine and adrenal androgen biosynthesis, such as the CYP17 inhibitors, abiraterone acetate and seviteronel, are also promising alternative treatments for AR-positive TNBC. In a phase II multicenter trial with metastatic or inoperable locally advanced AR-positive TNBC patients, abiraterone acetate (in combination with prednisone to offset aldosterone production) elicited a clinical benefit rate (CBR) of 20.0% at 6 months and median PFS at 2.8 months. Preliminary pharmacokinetic data from a phase I/II trial with seviteronel showed a significant reduction in testosterone in AR-positive TNBC patients [44]. Preclinical studies also demonstrate that seviteronel may sensitize AR-positive TNBC patients to radiotherapy [45]. Furthermore, since compensatory pathways often crosstalk with the AR pathway, the future of AR inhibition will likely require the inclusion of targeted therapies that impair these alternative pathways. Cyclin D1 and Rb protein expression is often upregulated in AR-positive TNBCs [46]. Thus, clinical trials are already underway that combine AR inhibitors with CDK4/6 inhibitors, such as palbociclib and ribociclib [44]. AR-positive TNBC biology is also characterized by PIK3CA mutations and p-AKT [23,47,48]. A multi-institutional phase I/II study (TBCRC032) has commenced to determine the safety and efficacy of combining AR inhibitors such as enzalutamide with the PI3K inhibitor, taselisib, in metastatic AR-positive BC patients [49]. The combination resulted in a significant increase in the CBR among AR-positive TNBC patients.

## 5. A Double-Edged Sword: Controversial Role of ARs in ER+ Breast Cancer and TNBC

Similar to ERs and PRs, the AR is a member of the nuclear steroid hormone receptor family and transcriptionally regulates target genes. Testosterone and dihydrotestosterone are androgens that directly or indirectly (as prohormones) stimulate AR-signaling [50,51]. Upon binding of androgens to an AR, the receptor translocates into the nucleus and binds to the promoter of target genes to enhance transcriptional activity [51]. AR-signaling plays an important role in both the development of normal breast tissue and in breast tumorigenesis and progression [52,53]. Several studies have defined androgens as potential tumor suppressors in ER-positive breast cancer with anti-proliferative activities. The anti-proliferative activity of ARs in ER-positive breast cancer is thought to be the result of crosstalk between AR and ER signaling pathways [54]. ERs promote proliferation by binding to estrogen response elements (EREs) in cis-regulatory elements of estrogen-regulated genes [51,55]. ARs can competitively bind to EREs to suppress estrogen-mediated tumor proliferation [52,56].

In contrast to ER-positive breast cancer, ARs promote proliferation of ER-/PR-negative breast cancer cell lines [57]. This finding is supported by studies by Garay et al. and Doanne et al. who raised the possibility of therapeutic targeting of the androgen pathway in TNBC [47,58]. Mechanistic studies in TNBC cell lines provide evidence that the AR interacts with AREs and stimulates tumor cell proliferation in an androgen-dependent manner [51].

Despite mechanistic studies in ER-/PR-negative cell lines, the role of AR signaling in TNBC is controversial [59,60,61]. AR expression in TNBC has been reported to range from as low as 7% to as high as 75% [61,62,63,64,65]. Studies investigating the prognostic role of the AR in TNBCs have similarly reported diverse results. Multiple groups have reported that negative AR status confers an aggressive disease course in TNBCs [12,63,64,66,67]. Loss of AR expression has been associated with younger age at presentation, lower stage, grade, mitotic scores, Ki-67, and lymphovascular invasion [60,61,63,68]. Additionally, several groups have reported that lack of AR expression in TNBC is associated with an increased risk of recurrence, distant metastasis, shorter DFS and shorter OS [63,69,70,71,72,73,74]. Paradoxically, some other studies have shown the opposite trend where AR positivity has been associated with younger age at diagnosis, higher nuclear grade, higher tumor stage, greater lymph node metastases and increased mortality [38,61,63,64,68,69,75,76,77].

The controversy over the AR is attributed to multiple factors, including variation in sample preservation (e.g., cold hypoxia time), use of different AR antibodies, staining methods, scoring methods, cut off values, lack of external validation, confounding effects of patient selection, and the existence of 15 different AR splice variants [78]. But perhaps one of the most significant contributors to this controversy is biogeographic ancestry. In a multi-institutional study, AR in TNBC was found to be a positive prognostic biomarker in US and Nigerian (West African) cohorts but a negative prognostic biomarker in women from Norway and Ireland [9]. This finding suggests that AR expression may confer a poor prognosis in TNBC that occurs in women of European ancestry but a favorable prognosis in women of West African ancestry. Differences in AR signaling networks in women of different genetic ancestry, however, is poorly understood.

## 6. QNBC—The Quadruple Threat

While some TNBC express AR, approximately 65–88% of TNBC lack AR expression. AR-negative TNBC is called quadruple-negative breast cancer (QNBC: ER^−^/PR^−^/HER2^−^/AR^−^) and is considered a “quadruple threat”. Accumulating evidence suggests that QNBCs are significantly more aggressive than AR-positive TNBCs. QNBC has also been linked to the clinically aggressive basal-like molecular phenotype; in contrast, AR-positive TNBCs are linked with a less aggressive luminal phenotype [10,13]. Thus, QNBC is increasingly recognized as an aggressive, hard-to-treat breast cancer subtype.

## 7. A New Racial Disparity in Breast Cancer: Characterization of QNBC Disparity in Women of African versus European Ancestry

Recent studies provide evidence that the AR is differentially expressed in TNBC from women of African- versus European-ancestry. Gasparini et al. showed that the frequency of AR-positive TNBC was greater in White/EA versus Black/AA women (25.5% versus 16.7%, respectively) [12]. In a US cohort, it was revealed that among Black/AA compared to White/EA TNBC patients, the percentage of patients negative for AR expression was significantly higher (80.1% vs. 70.3%, respectively) [9]. Davis et al. corroborated these findings by showing that in multiple publicly available cohorts, AR mRNA expression was lower in TNBCs from Black/AA versus White/EA women, irrespective of ER and PR status [10]. In the same study, 100% of Black/AA women with TNBC were shown to be AR-negative. Several groups have also reported that QNBC is even more prevalent among native West African than Black/AA women. In TNBC from East African and White/EA women, AR expression levels were similar [9,11,79]. These findings suggest that low AR expression in TNBC is strongly associated with West African-ancestry as opposed to East African- or European-ancestry.

Emerging evidence suggests that QNBC is clinically more aggressive in Black/AA versus White/EA women. PAM50 subtyping of AR-negative TNBC in TCGA, showed that Black/AAs have a higher percentage of basal-like tumors than White/EAs (77% versus 70%, respectively) [10]. In addition, subtyping of the same AR-negative tumors, showed that Black/AA women (compared to White/EA) have a higher percentage of aggressive TNBC subtypes such as BL1 (24% versus 16%), BL2 (16% versus 12%), and IM (24% versus 19%) subtypes but a lower percentage of the LAR (0% versus 2%) subtype [10,26]. Furthermore, among women with AR-negative basal-like TNBC, Black/AA women exhibited a significantly shorter time of progression than White/EA women [10].

## 8. Black/AA versus White/EA QNBC Biology: Novel Therapeutic Strategies to Reduce the Racially Disparate Burden in QNBC

Accumulating evidence suggests that QNBC is a highly aggressive breast cancer subtype and disproportionately impacts Black/AA women. Many groups have identified unique genes and pathways differentially regulated in QNBCs versus AR-positive TNBCs for targeted therapeutic intervention. Consistent with the basal-like phenotype associated with QNBCs, some studies have discovered that QNBCs are highly upregulated in proliferative markers such as EGFR, Ki-67, CDK6, and topoisomerase 2a (TOPO2A) compared to AR-positive TNBCs [12]. This evidence suggests QNBCs would be highly susceptible to inhibitors of proliferation such as EGFR, CDK6, and TOPO2A inhibitors. The cellular metabolic marker, long chain acyl-CoA synthetase 4 (ASCL4), was also found to be significantly upregulated in TNBCs absent of AR expression, suggesting ASCL4 inhibitors may be another viable therapeutic candidate [80]. Furthermore, as aligned with the highly immunogenic biology of QNBCs versus AR-positive TNBCs, QNBCs have also been found to be highly upregulated in T cell markers (CD4 and CD8), immune checkpoints (PD1, PD-L1, and CTLA4), and immune signaling pathways (ILR2B, CCR5, NFKBII2) compared to AR-enriched tumors [10]. These findings suggest that QNBCs may be highly sensitive to immunotherapeutic strategies, such as PD-L1 inhibitors.

However, with reports suggesting that QNBC is more aggressive in Black/AA than White/EA women, there is an urgent need for therapeutic targets specifically associated with QNBC disease in Black/AA women. Although limited, evidence suggests that differentially expressed genes and pathways in Black/AA versus White/EA QNBC exist as illustrated in Figure 1. Specifically, genes involved in immune cell signaling were found to be differentially expressed between Black/AA and White/EA women with QNBC, such as E2F1, NFKBIL2, CCL2, TGFB2, CEBPB, PDK1, IL12RB2, IL2RA, and SOS1 [10]. Furthermore, the CD4 T-cell marker and immune checkpoint genes PD-1, PD-L1, and CTLA-4, were found to be significantly upregulated in Black/AA compared to White/EA QNBC. Thus, QNBCs in Black/AA women may be highly susceptible to immunotherapeutic intervention, which could aid in reducing the racially disparate burden in QNBC.

## 9. Spotlight on KIFC1: A Promising Target for QNBC in Women of West African Descent

Kinesin family member C1 (KIFC1) is a microtubule binding protein that confers the survival of cancer cells with centrosome amplification, which is when a cell consists of more than 2 centrosomes per cell or has abnormally voluminous centrosomes [81,82]. Centrosome amplification is a hallmark of aggressive cancer as centrosome numbers are highly conserved in normal cells [83]. When cancer cells are burdened with extra centrosomes, they undergo erroneous multipolar spindle mitoses. These mitotic cell division errors can lead to improper segregation of chromosomes into each daughter cell, resulting in apoptotic cell death. KIFC1 blocks the apoptotic cell death program from occurring by clustering the extra centrosomes at opposite poles of the cell via crosslinking microtubules. This crosslinking allows cancer cells to undergo “pseudobipolar” spindle mitoses [82,84]. Thus, KIFC1 facilitates the persistence of genomically unstable cells and the expansion of aggressive cellular clones. Therefore, KIFC1 has emerged as a biomarker of aggressive breast cancer. Primary breast tumors overexpress KIFC1 relative to matched normal breast tissue, while KIFC1 is expressed higher in TNBCs compared to non-TNBCs [85,86]. Nuclear KIFC1 expression is associated with an advanced tumor grade as well as poorer OS and progression-free survival in breast cancer [86]. In TNBC, KIFC1 is among the top 1% of genes that are upregulated in TNBCs versus non-TNBCs and its overexpression is associated with a survival rate of <5 years [87]. Overexpression of KIFC1 in MDA-MB-231 and MDA-MB-468 TNBC cell lines conferred enhanced cell survival versus vector controls [88].

It was shown that nuclear KIFC1 expression is an independent biomarker of poor prognosis for Black/AA but not White/EA women with TNBC [89]. Furthermore, KIFC1 inhibition hindered proliferation and migration of Black/AA-derived TNBC cell lines to a greater extent compared to White/EA-derived TNBC cell lines [89]. These findings suggest that KIFC1 may be specifically critical for the progression of Black/AA TNBC biology and that targeting KIFC1 in Black/AA women may help reduce the racially disparate burden in TNBC. Since emerging evidence suggest that QNBCs are more aggressive in Black/AAs versus White/EAs, we hypothesize that KIFC1 may also be a promising target for Black/AA women with QNBC.

Furthermore, KIFC1 has been identified as the top hit among centrosome clustering genes in a genome-wide Drosophila screen, suggesting it may be the best centrosome clustering protein to target [82]. In TNBC, KIFC1 has also been identified as a malignant cell-specific dependency factor, which supports KIFC1 inhibition as a highly effective pharmacological strategy for aggressive TNBC [87]. Additionally, KIFC1 is non-essential for healthy cells with 2 centrosomes but indispensable for proper cell division of cancer cells with supernumerary centrosomes. Thus, KIFC1 inhibitors can selectively kill cancer cells burdened with extra centrosomes unlike standard chemotherapies [90]. Hence, KIFC1 inhibition may be a promising anti-cancer strategy for QNBC women of West African ancestry.

## 10. Black/AA versus White/EA QNBC Epigenomics: The Potential Role of Epigenetic Modifications in the Racially Disparate Burden in QNBC

Epigenetic modifications such as DNA methylation, microRNA (miRNA)-mediated gene silencing, histone posttranslational modifications and associated alterations in chromatin structure have long been shown to be key players in aggressive breast pathogenesis. An increasing body of evidence suggests that differential epigenetic patterns exhibited between Black/AA and White/EA women may be underlying the racially disparate burden in breast cancer [91,92,93,94,95]. Specifically, certain epigenetic markers have been linked to the predisposal of more aggressive breast cancer subtypes, such as TNBC and QNBC [96]. Recent evidence has identified distinct epigenomic profiles between TNBC and QNBC tumors as both DNA and RNA epigenetic modifications have been known to modulate androgen receptor expression. In particular, a molecular switch has been suggested to be responsible for the transition of TNBC to QNBC, or inactivation of the AR in TNBC, via hypermethylation of the AR promoter and/or miRNA-mediated transcriptional repression of the AR gene [97]. This transition has been suggested to promote more aggressive TNBC. These epigenetic changes can also alter genes involved in critical processes such as DNA repair, metabolism, and invasion/metastasis. Thus, differential epigenetic profiles could be conferring increased prevalence of QNBC and more aggressive QNBC disease in Black/AA compared to White/EA women.

Although limited, emerging evidence has begun to dissect the role of DNA methylation and miRNA-mediated gene regulation in conferring increased QNBC incidence and aggressiveness. Differential DNA methylation patterns in oncogenes and tumor suppressor genes among Black/AA and White/EA women have been observed to occur more frequently among ER-/PR-negative as opposed to ER-/PR-positive breast cancers, which could be underlying the increased incidence of TNBC and QNBC in Black/AA women [91,94]. miRNA regulatory networks have been suggested to alter the expression of genes in signaling pathways associated with QNBC and Black/AA QNBC, such as PD-1, PD-L1, CTLA-4, epidermal growth factor receptor (EGFR), phosphatase and tensin homolog (PTEN), ACSL4, S-phase kinase associated protein 2, and engrailed homobox1 [98]. Specifically, overexpression of miR-17-5p has been shown to downregulate PTEN, also suppressed in QNBC, via targeting the 3′-untranslated mRNA region [99]. Overexpression of ACSL4, observed in QNBC, has been discovered to occur via differential transcriptional mechanisms in the *ACSL4* gene and downregulation of miR-211-5p [100,101]. Overexpression of the EGFR, observed in QNBC, was associated with downregulated miR-133a [102]. Increased miR-135b expression was associated with AR-negativity in TNBC as well as a high proliferation index, which occurs in QNBC, and suggested to promote QNBC pathogenesis via targeting the TGF-β, WNT, and ERBB signaling pathways [103]. In an unpublished study, Angajala and colleagues discovered miRNAs differentially expressed between Black/AA and White/EA QNBC patients in The Cancer Genome Atlas (TCGA), which also correlated with AR expression, such as hsa-mir-135b, has-mir-18a, and has-mir-577, hsa-mir-500a, has-mir-181a-2, has-let-7d, has-mir-92a-2, has-mir-150, has-mir-17, hsa-mir-92a-1, has-mir-30a, has-mir-210, has-mir-455, hsa-mir-455, hsa-mir-130a, and has-mir-20a. The Women’s Circle of Health Study identified single nucleotide polymorphisms (SNPs) in miRNAs in Black/AA women that were associated with an increased risk for ER-negative and/or TNBC, specifically in hsa-mir-219, hsa-mir-595, hsa-mir-204, and hsa-mir-513a-2 [104]. Hence, epigenetic reprogramming via DNA methylation inhibitors or miRNA mimics/inhibitors in Black/AA women may help reduce QNBC incidence and aggressiveness in this patient subpopulation.

## 11. Black/AA versus White/EA QNBC Non-Biology: The Potential Role of Non-Genetic Risk Factors in the Racially Disparate Burden in QNBC

Disparities in non-genetic risk factors have long been reported to contribute to the gap in survival rates between Black/AA and White/EA women with breast cancer. These non-genetic risk factors include lifestyle (i.e., diet, physical activity, sleep), socioeconomic status (i.e., income level, education), access to quality oncological care (i.e., screening, health insurance, transportation), reproductive factors (i.e., parity, age at menarche, breastfeeding), anthropometrics (i.e., body mass index, waist/hip ratio), and comorbidities (i.e., obesity, diabetes, hypertension) [7,105,106,107]. Black/AA women experience disproportionately poorer access to adequate foods, physical activity, sleep patterns, breastfeeding, income levels, education, healthcare/screening, and health insurance compared to their White/EA counterparts [105,108]. Additionally, Black/AA women experience a greater burden of co-morbid diseases than White/EA women [109]. These unequal living standards and increased prevalence of co-morbid illnesses have also been suggested to underlie the acquisition of more aggressive subtypes, such as TNBC, in Black/AA versus White/EA women [7,105,108].

The Carolina Breast Cancer Study (CBCS) identified an association between increased waist/hip ratio and an increased risk of developing the basal-like subset of TNBC [110]. Other non-genetic risk factors such as co-morbid conditions, poor sleep patterns, lack of physical activity, stress, and lack of breast feeding have been linked to increased development of TNBC versus other subtypes via increased tumor-promoting inflammation. Specifically, poor sleep can suppress the immune system, cause alterations in the methylation of inflammatory genes such as IFN and tumor necrosis factor (TNF), as well as upregulate cancer-stimulatory cytokines [111,112]. Physical activity can boost the anti-tumoral immune response as well as prevent a high body mass index and waist/hip ratio [113]. A lack of breastfeeding postpartum can lead to the mammary gland undergoing involution, which can result in increased tissue inflammation [114,115]. Higher rates of residence in low SES neighborhoods among Black/AA communities may lead to increased psychological distress, which has been reported to elevate levels of inflammatory markers such as interleukin 6 (IL-6) [105,116]. Lastly, co-morbid diseases such as obesity, diabetes mellitus, and hypertension have all been reported to increase tissue inflammation. Obesity can upregulate circulating levels of insulin, inflammatory cytokines such as IL-6, IL-8, TNF, leptin, chemokines, and transforming growth factor beta (TGF-β) [105,117,118]. These inflammatory markers can activate signaling networks that promote cell proliferation and genomic instability, such as PI3K-AKT, signal transducer and activator of transcription 3 (STAT3), nuclear factor-ΚB (NFΚB), EZH2 and Wnt signaling. Diabetes and hypertension can also promote the upregulation of inflammatory cytokines [7,105]. Since the basal-like and immunogenic subsets of TNBC are reflective of QNBC and Black/AA QNBC biology, we postulate that these non-genetic risk factors may also play a role in the greater incidence of QNBC and more aggressive QNBC disease among Black/AA women compared to White/EA women as shown in Figure 2. Thus, further investigation into this link is warranted to adequately address the racially disparate burden in QNBC.

## 12. Conclusions and Future Perspectives

Addressing the racial disparity in the highly aggressive breast cancer subtype, QNBC, could significantly contribute to reducing the racially disparate burden in breast cancer. We assert that successfully addressing this subtype disparity would require a multifaceted approach. Particularly, the US and worldwide racial disparity in QNBC is still poorly defined and documented. We suggest large cancer statistics programs such as the Surveillance, Epidemiology, and End Results (SEER), Center for Disease Control and Prevention (CDC) as well as predominately Black/AA cohorts such as the CBCS and Women’s Circle of Health Study (WCHS) investigate current statistics on the prevalence and impact of QNBC. Within these studies, we recommend controlling for non-genetic confounding factors such as SES and access to care. We also suggest increasing the pool of Black/AA women in publicly available datasets to better define the impact of AR expression in TNBC tumors of West African descent, and thorough investigating potential molecular targets for racially distinct QNBC tumors. Furthermore, we propose increased investigation of AR IHC-based expression for identification of optimal staining methodologies and cut-off values to successfully incorporate AR evaluation into routine clinical practice for QNBC women of West African descent. We strongly advocate for high-granularity genetic ancestry typing in these studies to refine risk-stratification of subpopulations most susceptible to specific therapeutic strategies for QNBC.

In addition, we recommend increased research into the molecular and biological mechanisms underlying increased QNBC incidence and mortality among individuals of West African descent, as well as how AR signaling differs between TNBC from Black/AA and White/EA women for optimal therapeutic intervention. We propose investigations into the role of KIFC1 as a potential therapeutic target for QNBCs of West African ancestry. Our overview of preliminary evidence suggesting that epigenetic modifications may be predisposing Black/AA women to QNBC pathogenesis suggest increased investigation into the role of other epigenetic modifications in the racially disparate burden in QNBC as well as into novel treatment strategies targeting these modifications. Furthermore, we encourage addressing the previously mentioned non-genetic risk factors, such as lifestyle and SES, in order to adequately address this new clinical challenge.

## Figures and Tables

**Figure 1 cancers-14-04484-f001:**
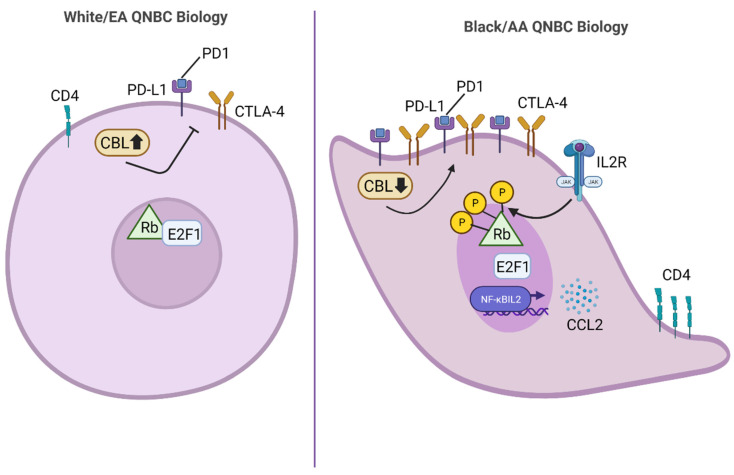
Distinctions in QNBC molecular cell biology among Black/AA and White/EA patients. Black/AA QNBCs have been shown to exhibit significantly upregulated levels of the T cell marker antigen, CD4, and immune checkpoint genes, PDL-1, PD1, CTLA-4 compared to White/EA QNBCs. In addition, Black/AA QNBCs have been reported to display significantly lower levels of CBL and significantly higher levels of E2F1, NFKBIL2, and CCL2 gene expression. CBL negatively regulates immune checkpoints; thus, downregulation of CBL in Black/AA QNBC cells can result in increased immune checkpoint activity. Observed upregulation of E2F1 in Black/AA QNBCs may be a result of increased IL2R signaling which can lead to hyperphosphorylation of Rb and thus, release of the E2F1 transcription factor. Upregulation of NFKBIL2 in Black/AA QNBCs can lead to increased transcription of cytokines and chemokines, such as CCL2. Abbreviations: PD-L1, programmed death-ligand; PD-1, programmed cell death protein 1; CTLA-4, cytotoxic T-lymphocyte-associated antigen 4; CBL, casitas B-lineage lymphoma; E2F1, E2 promoter binding factor 1; NFKBIL2, nuclear factor kappa B subunit 2; CCL2, chemokine (C-C motif) ligand 2; CD4, cluster of differentiation 4; IL2R, interleukin-2 receptor; JAK, Janus kinase; Rb, retinoblastoma.

**Figure 2 cancers-14-04484-f002:**
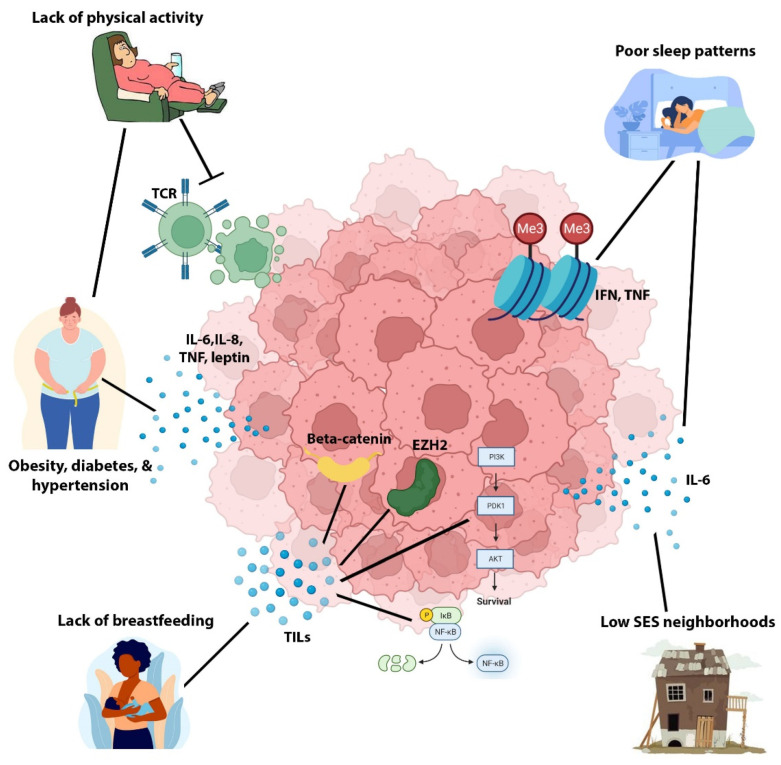
Potential influence of non-genetic risk factors on aggressive QNBC biology. Non-genetic risk factors such as co-morbid conditions (obesity, diabetes mellitus, and hypertension) poor sleep patterns, lack of physical activity, stress, low socioeconomic status (SES), and lack of breast feeding has been linked to aggressive basal-like TNBC biology, which reflects QNBC, via upregulating tumor-promoting inflammation and downregulating antitumor immunity. Abbreviations: IFN, interferon; TNF, tumor necrosis factor; IL-6, interleukin-6; IL-8, interleukin-8; TILs, tumor infiltrating lymphocytes; TCR, T cell receptor.

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
