# Peer review of "Racial Disparity in Quadruple Negative Breast Cancer: Aggressive Biology and Potential Therapeutic Targeting and Prevention"

_cancers, 2022, doi:10.3390/cancers14184484_

Round 1
Reviewer 1 Report
Thank you for the opportunity to review this manuscript. The authors have provided a very thorough and well-written review of the current state of the literature surrounding QNBC and known racial disparities between White and Black women. The literature and accompanying summary is scientifically sound and presented outstandingly. If possible, my only comment would be that the authors perhaps include one small paragraph describing known disparities among other racial/ethnic groups. This can be a short paragraph just describing whether these disparities are unique to Black vs White women or beyond that.
Author Response
Response to reviewers
Reviewer 1:
Thank you for the opportunity to review this manuscript. The authors have provided a very thorough and well-written review of the current state of the literature surrounding QNBC and known racial disparities between White and Black women. The literature and accompanying summary is scientifically sound and presented outstandingly. If possible, my only comment would be that the authors perhaps include one small paragraph describing known disparities among other racial/ethnic groups. This can be a short paragraph just describing whether these disparities are unique to Black vs White women or beyond that.
Response: Great suggestion! We have included new paragraph (second paragraph) of the introduction section to include these statistics.
Reviewer 2 Report
The authors review disparities in breast cancer outcomes for patients with an afroamerican origin, considering the higher incidence of TNBC and more recently classification of their disease within the QNBC context. They discuss racial disparities in understanding the biomolecular basis of breast cancer in this patient group. The work presented by the authors is timely and highly relevant, and I am in agreement that this is an important area of cancer research requiring extensive research to improve patient prognosis.
My specific comments relating to this manuscript are included as follows;
1) On the first page of the article, it would be good to present some statistics on the number of patients from the AA community impacted by breast ca, the incidence of TNBC in this patient group and survival outcomes, where the information is published.
2) Section 3- AR as a therapeutic target appears a little too early in the manuscript- the biomolecular basis needs to come first, then this topic can be later covered, including future prospects in targeting AR
3) Section 4- AR-negative expression statistics are presented- Is the proportion of AA patients with breast cancer impacted by AR negative QNBC known? This is later covered in section 6- I think a little rearrangement of the sections would help avoid confusion and improve clarity.
4) The biomolecular basis of AR is not extensively explained in the context of tumour aggressiveness and biomolecular pathways impacted by its absence of expression.
5) Section 7- the review presents the rationale for targeting AR in QNBC for AA patients with breast cancer. Later in section 7, the authors refer to limited evidence of differentially expressed genes and pathways in AA vs EA patients with breast cancer. Is this a consequence of lack data/studies comparing patient cohorts or an actual lack of difference? This could be further clarified? Is there an immune cell signalling role for AR? A better link could be made between these concepts.
6) What is more broadly known about the interplay between AR and KIFC?
7) Have any multiomics studies been performed to study the association between AR expression, and the proteome and metabolome? Many biorepositories with cell line and patient clinical data will contain these information?
8) Figure 2 shows the non-genetic risk factors for QNBC biology. Are these established in the literature from clinical studies of biomarkers or primarily obtained from pharmacoepidemiology studies?
Author Response
Reviewer 2:
The authors review disparities in breast cancer outcomes for patients with an afroamerican origin, considering the higher incidence of TNBC and more recently classification of their disease within the QNBC context. They discuss racial disparities in understanding the biomolecular basis of breast cancer in this patient group. The work presented by the authors is timely and highly relevant, and I am in agreement that this is an important area of cancer research requiring extensive research to improve patient prognosis.
Response: We are grateful for your kind and insightful comments. Please find our point-by-point responses to your questions and suggestions below:
My specific comments relating to this manuscript are included as follows;
1) On the first page of the article, it would be good to present some statistics on the number of patients from the AA community impacted by breast ca, the incidence of TNBC in this patient group and survival outcomes, where the information is published.
Response: Great suggestion. We have updated the second paragraph of the introduction section with these pertinent statistics and accompanying references.
2) Section 3- AR as a therapeutic target appears a little too early in the manuscript- the biomolecular basis needs to come first, then this topic can be later covered, including future prospects in targeting AR
Response: Great point. We have incorporated a new section (3) entitled, “AR signaling/pathway” in the manuscript that describes the biomolecular basis of AR. We have also included a second paragraph in section 4 (AR pathway as a therapeutic target for TNBC) that discusses future prospects in targeting AR in TNBC.
3) Section 4- AR-negative expression statistics are presented- Is the proportion of AA patients with breast cancer impacted by AR negative QNBC known? This is later covered in section 6- I think a little rearrangement of the sections would help avoid confusion and improve clarity.
Response: We strongly agree. We moved the “QNBC-the quadruple threat” (now section 6) to below the “A double-edged sword: Controversial role of AR in ER+ breast cancer and TNBC” (section 5) to improve organization and flow. We have also included an additional statistic on the percentage of AA patients negative for AR versus EA patients in section 7.
4) The biomolecular basis of AR is not extensively explained in the context of tumour aggressiveness and biomolecular pathways impacted by its absence of expression.
Response: We strongly agree. We moved the “QNBC-the quadruple threat” (now section 6) to below the “A double-edged sword: Controversial role of AR in ER+ breast cancer and TNBC” (section 5) to improve organization and flow. We have also included an additional statistic on the percentage of AA patients negative for AR versus EA patients in the first paragraph of section 7.
5) Section 7- the review presents the rationale for targeting AR in QNBC for AA patients with breast cancer. Later in section 7, the authors refer to limited evidence of differentially expressed genes and pathways in AA vs EA patients with breast cancer. Is this a consequence of lack data/studies comparing patient cohorts or an actual lack of difference? This could be further clarified? Is there an immune cell signalling role for AR? A better link could be made between these concepts.
Response: Discussion of AR as a therapeutic target in AR-positive TNBC was discussed in section 4 to show that AR is only targetable and actionable in this subgroup of TNBCs. However, this subgroup of TNBCs is less prevalent in Black/AA women suggesting that alternative therapeutic strategies are urgently needed to address the predominant QNBC (AR-negative TNBC) subtype observed among this population, which is discussed in section 8. There is limited evidence of molecular differences between AR-positive vs AR-negative TNBC as well as between AA and EA QNBC biology. We provide known differences and suggest further investigation into more distinctions to mitigate the overall racially disparate burden in breast cancer. In section 8, we discuss how QNBC or lack of AR signaling is associated with upregulation in immune cell markers, checkpoints, and pathways suggesting immunotherapeutic intervention may be a promising strategy for AA QNBCs.
6) What is more broadly known about the interplay between AR and KIFC?
Response: Not much is known about the interplay between AR and KIFC1 other than the links we provide in section 9 of the article. However, our group is currently investigating this link in the laboratory and we plan to publish our data and findings in the near future.
7) Have any multiomics studies been performed to study the association between AR expression, and the proteome and metabolome? Many biorepositories with cell line and patient clinical data will contain these information?
Response: Great ideas for future investigation! These associations have not been published yet, however future studies should perform these multi-omic analyses to gain a better understanding of the complex biomolecular role of AR signaling in TNBC for therapeutic intervention.
8) Figure 2 shows the non-genetic risk factors for QNBC biology. Are these established in the literature from clinical studies of biomarkers or primarily obtained from pharmacoepidemiology studies?
Response: Great question. These non-genetic risk factors have been established in robust cancer epidemiology studies published over the past several decades.
Round 2
Reviewer 2 Report
The authors have responded to all my comments and adequately addressed them- Excellent work!